# A Review of Keratoconus Cross-Linking Treatment Methods

**DOI:** 10.3390/jcm14051702

**Published:** 2025-03-03

**Authors:** Natalie Papachristoforou, Anthony Ueno, Kamila Ledwos, Jerzy Bartuś, Anna Nowińska, Izabella Karska-Basta

**Affiliations:** 1Ministry Hospital of Internal Affairs and Administration, 30-053 Krakow, Poland; natalienicole120@gmail.com (N.P.); ueno.anthony@gmail.com (A.U.); jerzy.bartus@gmail.com (J.B.); 2Stefan Zeromski Specialist Hospital, Independent Public Healthcare Institution, 31-913 Krakow, Poland; 3Clinical Department of Ophthalmology, Faculty of Medical Sciences in Zabrze, Medical University of Silesia, 40-055 Katowice, Poland; 4Clinic of Ophthalmology and Ocular Oncology, University Hospital, 30-688 Krakow, Poland; ikarska@su.krakow.pl; 5Faculty of Medicine, Department of Ophthalmology and Ocular Oncology, Jagiellonian University Medical College, 31-008 Krakow, Poland

**Keywords:** keratoconus, cross-linking, cornea

## Abstract

Corneal collagen cross-linking (CXL) is a therapeutic intervention that utilizes riboflavin photochemical activation with ultraviolet-A (UV-A) light to induce covalent cross-links within the stromal corneal fibers, effectively increasing corneal biomechanical stability and halting the progressive ectasia. The method was introduced in the late 1990s in Germany at the University of Dresden. The cross-linking method using the Avedro system (Waltham, MA, USA) was approved by the US Food and Drug Administration (FDA) on 18 April 2016, based on three prospective, multicenter, randomized clinical trials for keratoconus and other corneal ectasias. Recent innovations in CXL include a range of new treatment protocols and methods, which have been introduced to further enhance the clinical effectiveness, efficiency, and safety of CXL. These modifications encompass approaches like transepithelial or epithelium-on CXL (TE-CXL or epi-on CXL), accelerated CXL (ACXL), pulsed CXL (PL-CXL), transepithelial iontophoresis-assisted crosslinking (I-CXL), diluted alcohol and iontophoresis-assisted corneal cross-linking (DAI-CXL), slit-lamp CXL, and CXL plus (combined) methods. This review synthesizes findings on currently used modifications of the cross-linking method, the effectiveness, and directions of development of this currently dominant surgical method of treating corneal ectasia. This review concentrates on the long-term follow-up data, based on publications ranging from 1998 up to 2023.

## 1. Introduction

Keratoconus (KC) is a noninflammatory bilateral condition of the cornea, characterized by thinning and a conical shape with bulging of the cornea. The cause is unknown, but the disease is often associated with other conditions such as neurodermatitis, allergic asthma, allergic rhinitis, Marfan syndrome, Ehlers–Danlos syndrome, and Turner and Down syndrome. In a relatively small percentage of cases (10%), there is a positive family history of KC. Clinical manifestations typically start in one eye but eventually affect both eyes. Symptoms include a progressive decrease in visual acuity, myopia, astigmatism, and photophobia [1]. The diagnosis of KC involves a slit lamp examination to observe the protrusion and thinning of the cornea, pachymetry to measure corneal thickness, and mainly computerized corneal topography to asses altered keratometry readings [1,2].

The epidemiology of KC varies greatly in different regions of the world. Moreover, the data are selective because research has so far been carried out in only a few countries. The prevalence varies greatly from 0.2 to 4790 per 100,000 people and the incidence ranges from 1.5 to 25 new cases per 100,000/year. The above-mentioned indicators occur mainly in people aged 20–30 [3,4,5].

For many years, the management of KC was mainly confined to the use of eyeglasses and rigid gas permeable contact lenses. Currently, in cases of progressive KC, corneal cross-linking (CXL) is a dominant surgical method of treatment. In advanced forms, especially in the case of corneal scarring, keratoplasty (corneal transplant) may be necessary. It should be clearly emphasized that since the introduction of the CXL method in the treatment of progressive KC at an early stage, the percentage of patients requiring corneal transplantation for this reason has significantly decreased [1].

CXL was initially introduced approximately 25 years ago as a means to prevent the progression of KC. In recent times, a minimally invasive CXL has emerged as a viable option for halting the progression of the disease [4]. The procedure is used to strengthen the corneal tissue. The principles of CXL involve the use of riboflavin (vitamin B2) as a photosensitizer and ultraviolet-A light (UV-A) to slow down or prevent the progression of KC [6]. The objective of CXL is to create strong covalent bonds within the corneal stroma, increasing its stiffness and resistance to KC progression. The Dresden protocol was named after the University of Dresden, where the method originated and was formulated in the late 1990s. A groundbreaking and minimally invasive method was introduced by Wollensak et al. in 2003 [7]. It is the revised version of the Dresden protocol, which demonstrated encouraging clinical outcomes. The results revealed a halt in the progression of the condition after the treatment and a diminished necessity for keratoplasty. Following this, a range of modified treatment protocols and methods have been introduced to further enhance the clinical effectiveness, efficiency, and safety of CXL. These modifications encompass approaches like transepithelial or epithelium-on CXL (TE-CXL or epi-on CXL), accelerated CXL (ACXL), pulsed CXL (PL-CXL), and numerous others, such as transepithelial iontophoresis-assisted crosslinking (I-CXL), diluted alcohol and iontophoresis-assisted corneal cross-linking (DAI-CXL), slit-lamp CXL, or CXL plus (combined) methods.

The following article is a review and analysis of the currently used modifications of the CXL method, the effectiveness, and directions of development of this currently dominant surgical method of treating KC. New protocols are continuously being developed, yet they are tested on relatively small groups with a short follow-up period. Our paper concentrates on methods currently used with long-term follow-up data, based on publications ranging from 1998 up to 2023.

## 2. Materials and Methods

The objective of this review was twofold: firstly, to provide a concise overview of KC and its primary treatment, CXL, elucidating the functional mechanisms of this procedure and tracing the evolution of the standard protocol, known as the Dresden protocol. Secondly, the aim was to delineate the progression of diverse modifications to the CXL method since the establishment of the Dresden protocol. This encompassed an exploration of the distinctions, advantages, and potential risks associated with these novel techniques. To achieve this, our literature search was deliberately expansive, targeting a diverse array of protocols with a broad international spectrum. Particular emphasis was placed on recent publications to accurately depict the current state of the art, with over half of the articles sourced originating from recent years.

The literature search was executed, employing the PubMed, Cochrane Library, and Google Scholar databases. The search was driven by a set of carefully selected keywords including “corneal cross-linking”, “keratoconus”, “the Dresden protocol”, “modified cross-linking”, “accelerated methods”, “transepithelial iontophoresis-assisted cross-linking”, “diluted alcohol and iontophoresis-assisted corneal cross-linking”, “pulsed methods”, “slit-lamp cross-linking”, and “combined method”. Records were excluded based on their title, abstract, and open access screening according to the following criteria: publications older than 10 years (unless relevant to the review) or lack of open access unless they were accessible via institutional licenses. Duplicate records were also removed, resulting in a final selection of 75 articles for analysis. The flow chart for study selection is presented in Figure 1. The Table 1 presents thegeneral summary of corneal cross-linking methods and their clinical outcomes. The table summarizing all the papers included in this review with key parameters is provided in the Appendix A.

## 3. Dresden Protocol CXL

The Dresden protocol process involves the use of ultraviolet radiation and a photosensitizing substance, most often riboflavin (vitamin B2) 0.1% in 20% dextran applied for 30 min. At the start of the procedure, the central part of a 7 mm area of corneal epithelium is gently rubbed until it is removed to facilitate the solution’s penetration. Then, drops of riboflavin are applied to the surface of the cornea, which is activated by exposure to UV-A light for 30 min for a total surface energy exposure of 5.4 J/cm^2^ (3 mW/cm^2^). UV radiation causes a chemical reaction between riboflavin and collagen in the cornea, leading to the formation of cross-links between the collagen fibers. These bonds strengthen and stabilize the structure of the cornea, reducing its tendency to further deform and deteriorate vision. The first prospective, nonrandomized clinical pilot study of CXL was performed by Wollensak et al. in 2003 [7]. The study involved twenty-three eyes of 22 patients with moderate or advanced progressive KC. As a result, 70% of eyes had regression of the disease with a reduction in the maximal keratometry (Kmax) readings by 2.01 diopters (D) and in the refractive error by 1.14 D. The Dresden protocol remains the gold standard for treating progressive KC, for this reason it is also called the standard protocol or conventional CXL (CCXL) [8]. For safety reasons and protection of corneal endothelial cells from potentially dangerous levels of UV-A exposure, corneas 400 μm or thicker can be qualified for the treatment. The results of long-term observations of the safety and effectiveness of the Dresden protocol are available. In 2015, Raiskup F. et al. published 10-year results of the method based on the Dresden protocol. The authors confirmed the safety of the method, the long-term result of reducing keratometric values (from 61.5 D to 55.3 D), and improving corrected distance visual acuity (CDVA) (0.14 logMAR units; approximately 1.4 lines on a standard Snellen chart) [9]. Poli M. et al., based on a 6-year study, observed inhibition of KC progression in 89% of patients after the procedure, while O’Brart DP. et al. did not observe disease progression in any of the patients included in the study during a 7-year follow-up [10,11]. Eslami M. et al. published a study with the results including 150 eyes of 112 patients with KC who underwent CXL with a minimum follow-up of 5 years. At the last follow-up visit, an improvement in CDVA, spherical and cylindrical refraction, average and steepest keratometry, and corneal aberrations were observed. At the last visit, 49 and 31 eyes had an improvement beyond 1 D in their spherical and cylindrical power, respectively, and 43 eyes had a flattening of their steepest keratometry. Progressive improvement over time was observed for spherical refraction, Kmax and Kmean, as well as root mean square (RMS) values of corneal high-order aberrations (HOAs), total, high, coma, and spherical aberrations [12]. More severe disease at the baseline correlated with an improvement in corneal aberrations over time. In the study by Iqbal M., in a group of 49 eyes, 40 eyes (81.6%) achieved a postoperative spherical equivalent (SE) refraction better than the attempted refraction. Both Kmax and SE refraction showed statistically significant and stable improvement from preoperative 51.95 ± 1.90 D and −7.90 ± 3.14 D to postoperative 50.19 ± 1.96 D and −6.35 ± 2.49 D, respectively. Two eyes (4%) showed KC progression at the end of 5th follow-up year [13].

CCXL was the first protocol implemented in the treatment of KC and other ectatic disorders. These include pellucid marginal degeneration (PMD), post-refractive surgery ectasia, keratoglobus, Terrien’s marginal degeneration, iatrogenic ectasia, and selected corneal dystrophies, such as lattice corneal dystrophy. Its development established the foundation for CXL as a safe, effective, and minimally invasive treatment. Its use has been validated across a wide range of populations, clinical settings, and stages of KC in Europe: Germany (Wollensak et al. [7]), Italy (Caporossi et al. [10,14]), France (Poli et al. [11]), and the UK (O’Brart et al. [11], Chowdhury et al. [15] KERALINK); North America: the United States (Hersh et al. [16,17]) and Canada (Legare et al. [18]); the Middle East: Saudi Arabia (Khattak et al. [19]) and Iran (Hashemi et al. [20]); Asia: India (Sachdev et al. [21]) and China (Gu et al. [22]); Australia (Wittig-Silva et al. [23]); Africa (Mark et al. [24]); South America: Brazil (Gadelha et al. [25]).

Other methods and modifications of CXL are commonly compared to the principles and outcomes established by the Dresden protocol, which serves as a benchmark for evaluating the efficacy, safety, and biomechanical effects of alternative approaches.

## 4. Modified CXL—General Considerations

Advancements in CXL have led to several modified techniques, namely accelerated CXL (ACXL), transepithelial or epithelium-on CXL (TE-CXL or epi-on CXL), pulsed CXL (PL-CXL), transepithelial iontophoresis-assisted crosslinking (I-CXL), diluted alcohol and iontophoresis-assisted corneal cross-linking (DAI-CXL), slit-lamp CXL, or CXL plus (combined) methods.

ACXL is a modified version of the conventional procedure. The objective of ACXL is to achieve the same therapeutic outcomes as the standard procedure but within a briefer treatment duration. In ACXL, higher levels of energy are applied for a reduced timeframe, usually ranging from 3 to 10 min. This abbreviated treatment period is achieved by elevating the intensity of UV-A light while upholding the effectiveness of the cross-linking process. This innovation allows for a substantial reduction in treatment time, condensing it from the customary 30 min session to as little as 3 to 10 min, all while maintaining the overall radiant exposure [26]. The shortened treatment duration can result in enhanced patient comfort, improved compliance and possibly reduced risk of infection [13]. This aspect is particularly advantageous for patients who might find the standard half-hour treatment uncomfortable or challenging to endure. Furthermore, the reduced treatment time has the potential to enhance the feasibility of performing CXL procedures in clinical settings. However, the suitability of ACXL varies based on factors like the severity of the condition, thickness of the cornea, and individual patient characteristics. Clearly defined criteria for patient selection are essential to ensure optimal treatment outcomes. Just like any novel medical technique, ongoing research, and clinical trials are being conducted to assess the safety and effectiveness of ACXL. Recent studies suggest that the use of ACXL treatment (employing an irradiance of 9 mW/cm^2^ for 10 min) presents a viable approach for managing mild to moderate KC. This marks a noteworthy advancement in refining CXL treatment and may also have implications for the treatment of other conditions like bullous keratopathy, and infectious keratitis, where CXL is also employed [6,26].

The necessity for epithelial surgical removal is a major contributor to post-operative CCXL complications, such as infective keratitis and abnormal wound healing. This has perpetuated interest in developing epi-on CXL or TE-CXL techniques. TE-CXL was developed as a modified procedure in 2004. Epi-on CXL is a modified method that aims to perform CXL without removing the epithelium, which could potentially lead to faster healing and reduced discomfort. The challenge with this method is that the epithelium acts as a barrier, making it harder for the riboflavin to penetrate deeply into the cornea. It utilizes different formulations and delivery methods of riboflavin to avoid epithelium removal. The diffusion of riboflavin can be achieved by several techniques, including modifying corneal epithelial permeability, changing the physicochemical properties of the riboflavin molecule, and direct delivery of the riboflavin molecule into the corneal stroma by the creation of an epithelial flap or pocket. Therefore, different techniques, such as iontophoresis (using an electric current to facilitate riboflavin penetration), have been explored to enhance the efficacy of epi-on CXL. Innovative approaches such as epi-flap CXL have also been described, which is shown to be associated with less postoperative pain and anterior stromal haze when compared to conventional epithelium-off CXL [27,28,29]. Studies have shown that TE-CXL is less effective in halting the progression of KC than CCXL, but was associated with significantly fewer complications than conventional epithelium-off CXL (2% vs. 4%, respectively) [30].

Another surgical approach of modified CXL is PL-CXL. In this method, pulsing the UV-A light during the crosslinking process is theoretically aimed at restarting the photodynamic type 2 reaction. This allows for the replenishment of oxygen in the stromal tissue, leading to increased release of singlet oxygen and enhanced crosslinking of collagen molecules. The concept behind pulsing UV-A irradiation is that it could potentially result in fewer adverse effects even at higher irradiation levels, all while promoting a more efficient crosslinking procedure. The effectiveness of using short pulsing durations is uncertain, and there is a need for further refinement of the protocol. Additionally, the fact that pulsed ACXL (pl-ACXL) takes twice as long to complete compared to continuous ACXL (c-ACXL) without any evident clinical advantage should be taken into careful consideration, especially in countries where KC is widespread, and healthcare resources are in high demand. CXL assisted by transepithelial iontophoresis (I-CXL) is a technique that involves the use of electric current to aid in the penetration of riboflavin through the corneal epithelium. This solution may comprise 0.1% riboflavin and lacks dextran or sodium chloride. However, it may include various enhancers (like benzalkonium chloride, high-concentration sodium chloride, or sodium ethylene diamine tetra acetic acid) to aid in the penetration through the corneal epithelium. A corneal ring electrode is positioned on the cornea, containing 0.5 mL of riboflavin solution. It is then linked to a continuous current generator, which facilitates the transfer of riboflavin through the epithelium within a span of 5 min. Subsequently, the cornea undergoes UV-A irradiation following the established protocols for ACXL. This method aims to enhance the delivery of riboflavin into the corneal stroma, allowing for more efficient and effective cross-linking of collagen fibers. The use of iontophoresis in conjunction with CXL is intended to optimize the treatment process. However, the comprehensive effectiveness of I-CXL should be approached with care, considering notable differences in both clinical and methodological aspects among the studies being compared [30]. In 2017, Bilgihan K. et al. introduced a modified version of I-CXL known as transepithelial DAI-CXL. This innovative cross-linking protocol incorporated two enhancers: a diluted alcohol solution containing 10% ethanol and iontophoresis [31].

Innovative approaches, such as combining CXL with intracorneal ring segment implantation or topography-guided photorefractive keratectomy (PRK), have shown promise in managing advanced keratoconus. The synergistic effect of these techniques may help reshape the cornea while simultaneously strengthening its biomechanical properties through cross-linking. Further research assessing this dual approach is needed for the widespread adoption of these methods into clinical practice [32,33,34,35]. Additionally, slit-lamp delivery methods for UV-A irradiation are being explored to improve accessibility in clinical settings by allowing the procedure to be performed without requiring specialized surgical suites. These approaches may reduce the procedural burden and cost while expanding treatment availability to underserved populations [36].

## 5. Accelerated Methods

ACXL protocols leverage the advantages of the Bunson–Roscoe law of photochemical reciprocity to reduce irradiation time by increasing the irradiation intensity while maintaining a constant cumulative dose of energy applied to the cornea [37]. Conversely, the CCXL protocol uses a lower irradiation intensity of 3 mW/cm^2^ and an irradiation time of 30 min [7]. However, high-energy settings of up to 43 mW/cm^2^ and in ex vivo studies even 45 mW/cm^2^ have been developed, reducing the irradiation time to 2 and 1 min, respectively, and shortening the soak time of the riboflavin solution [38,39]. It is assumed that the cumulative dose of irradiation should not exceed 5.4 J/cm^2^ to minimize the risk of complications. On balance, elevated energy levels were implemented to shorten the duration of CXL and accelerate the process. While accelerated protocols offer the clear benefits of shorter treatment duration, reduced patient discomfort, and decreased risk of postoperative complications, as well as increased cost-effectiveness, the clinical advantages are still being debated in several studies. Although no health economics studies confirming cost-effectiveness have been published yet, several studies have investigated clinical effectiveness.

In the Brittingham S. et al. study, the rapid protocol was compared to the Dresden protocol. A total of 131 eyes were analyzed, 81 eyes in the Dresden protocol group and 50 eyes in the rapid protocol group. The demarcation line was revealed in 76.5% (62/81 eyes) of the treated corneas in the standard protocol group, whereas such a demarcation line was observed in only 22% (11/50) of eyes treated with the rapid protocol. The demarcation line was significantly more superficial in the rapid protocol group. Corneal topography values between baseline and 12 months after CXL showed a mean change of 0.76 D in Kmax (SD ± 2.7) in the standard protocol group versus a mean change of +0.72 D in Kmax (SD ± 1.5) in the rapid protocol [40].

In a recent study by Badawi AE. et al., CCXL was compared with TE-CXL and ACXL protocols in a cohort of 104 eyes from 53 patients. The study examined differences in densitometry, CDVA, and corneal haze among the groups. The results showed that the ACXL group had a longer recovery time from corneal haze to reach baseline levels, as well as increased persistent corneal densitometry, particularly in the anterior 120 µm, one year after surgery. This increase in densitometry may be attributed to differences in soaking durations and concentrations of riboflavin solutions used in the protocols. In particular, the higher concentration of riboflavin solution in the anterior corneal stroma of the ACXL group, compared to the CCXL group, may have contributed to this effect. Another contributing factor may be the concentration of oxygen within the corneal stroma, as the ACXL protocol may not allow enough time for oxygen diffusion. During a 12-month follow-up, no significant correlations were found between changes in CDVA and densitometry over time. Furthermore, differences in corneal haze did not have a significant effect on postoperative CDVA [41].

In a retrospective study based on a prospectively built database, Wajnsztajn D. et al. analyzed postoperative outcomes in a large patient population of 613 eyes that underwent surgery over 11 years with at least 1-year follow-up. The study compared various accelerated and non-accelerated protocols, as well as epithelium-off and epithelium-on approaches. The results showed that non-accelerated epithelium-off protocols achieved a better outcome in terms of greater corneal flattening, which confirms the findings of two other meta-analyses that included a total of 33 trials. Interestingly, the study did not find gender or adult age to be predictors of the efficiency of corneal flattening [42].

In a prospective study by Tian M. et al., a potential factor that could predict the possibility of stabilizing progression or improving corneal morphology after corneal crosslinking was explored. The study found that thinner preoperative central corneal thickness (CCT) or greater Kmax values were associated with a higher likelihood of flattening more than 1D in Kmax values after TE-ACXL. This effect can be attributed to the fact that thinner corneas provide better infiltration for riboflavin into the deeper corneal stroma, while thicker corneas may limit the depth of riboflavin infiltration and thereby affect the effect of CXL. Notably, the study did not find a correlation between age and postoperative changes in Kmax, which the authors explained by the young age of their patient population (under 30 years) since the corneal morphology of adolescents is easier to change [43].

Aldairi et al. conducted a study to compare the short-term outcomes of the conventional Dresden protocol and an accelerated protocol for halting KC progression. The results demonstrated that both methods were safe and efficient, and the final CDVA was similar in both groups after a 9-month follow-up period, with the last follow-up visit conducted after up to 18 months. The accelerated protocol employed in the study utilized a higher total irradiance of 7.2 J/cm^2^. Notably, Kmean values regressed in both groups as early as 3 months, but despite the fact that the AXCL group had steeper keratometry values, regression of Kmean, flat and steep K values surpassed those of CCXL at the last visit. However, the decrease in corneal astigmatism was significant in the conventional CXL group, unlike the AXCL group [44].

A study conducted by Salman M. et al. aimed to compare the conventional Dresden protocol with an accelerated protocol (10 mW/cm^2^, 9 min) performed using an epithelium-off approach. The study evaluated preoperative and 18–30 months postoperative CDVA, refraction, corneal topography, corneal tomography, and anterior and posterior HOAs. The CCXL group showed significantly higher anterior corneal flattening, more increase in posterior steepening, further decrease in posterior astigmatism, and more reduction in the minimum thickness compared to the ACXL group. Nonetheless, both protocols resulted in postoperative improvements in CDVA and corneal HOAs, with the ACXL protocol showing a significantly higher improvement in CDVA [45]. In contrast, a meta-analysis conducted by Kobashi H. and Tsubota K. reported that while overall ACXL and CCXL protocols have similar outcomes, the CCXL approach demonstrated superiority over ACXL by exhibiting CDVA after 1-year follow-up. However, the authors acknowledged that the observed difference may not be clinically significant [46].

The safety and efficacy of various CXL protocols were the subject of a study conducted by Singh T. et al. The study compared the results of the conventional Dresden protocol with hypotonic solution treatment for thin corneas and ACXL using a 5 min exposure to 18 mW/cm^2^. At the end of the 12-month follow-up period, the results showed that the accelerated protocol was safe and as efficient as the conventional CXL, as small differences in Kmean value and endothelial count were not statistically significant [47]. These findings are consistent with several other studies, such as Razmjoo H. et al., who favor the ACXL method due to its better workflow fluency and shorter surgery duration [48].

In pediatric populations, CCXL has been found to be superior to ACXL, according to Iqbal M. et al. During the first year after treatment, both groups showed improvement, but after 2 years, the CCXL group (91 eyes) continued to improve, while the ACXL group (92 eyes) regressed, leading to significant differences in all visual, refractive, and keratometric components [49]. However, a meta-analysis by Fard A. et al. comparing 28 studies, which included 1300 eyes with CCXL, ACXL, and transepithelial approaches, found comparable results between conventional and accelerated protocols. Thus, ACXL is considered the preferred technique in pediatric patients, owing to its reduced surgery time [50]. One possible factor that could influence the varying efficacy between CCXL and AXCL in several studies is the depth of the demarcation line, which serves as a surrogate indicator for treatment depth and the associated impact on corneal biomechanics after CXL [45]. Several studies have investigated the depth of the demarcation line to assess the impact of CXL. Most researchers agree that the depth of the demarcation line is shallower after ACXL than after CCXL, likely due to the shorter soaking time with ACXL [49]. However, all of the aforementioned studies examining accelerated protocols have evaluated outcomes with follow-up periods of up to 2 years. As Kobashi H. and Tsubota K. note in their meta-analysis, longer-term results with extended follow-up periods are necessary to more comprehensively evaluate the clinical efficacy of the two procedures. Differences in clinical efficacy may still be revealed with longer follow-up periods, particularly in light of the divergent depth of the demarcation line observed after ACXL and CCXL [46].

## 6. Transepithelial Iontophoresis-Assisted Crosslinking (I-CXL)

Transepithelial iontophoresis is a noninvasive method used in KC treatment, where electromigration is used to facilitate the penetration of an ionized substance into the tissue through the tight junctions of an intact corneal epithelium [37,51]; it is a modern technique that is carried out using topical anesthesia [52,53]. The I-CXL procedure relies on the generation of repulsive electromotive forces through the application of a low electrical current in an iontophoretic chamber which enhances the penetration of riboflavin into the corneal stroma. I-CXL is a therapy that relies on the use of riboflavin, a small molecule that plays a critical role in the procedure. This molecule has a molecular weight of 376.4 g/mol and a negative charge, which makes it ideal for the treatment. The specific formula of riboflavin used in the procedure surpasses methods that utilize topical riboflavin application due to its ability to leave the epithelium intact and shorten the riboflavin saturation phase. The preclinical trials have demonstrated that I-CXL has a beneficial effect on the mechanical strength of the cornea [54]. During the I-CXL procedure, a solution of 0.1% riboflavin is used without any additional dextran or sodium chloride. Additionally, various enhancers such as benzalkonium chloride, sodium chloride, sodium ethylene diamine tetra acetic acid, or trometamol may be applied to facilitate higher penetration across the epithelium [52]. The I-CXL procedure requires an electrical circuit that comprises a current generator and two electrodes. The first electrode, which is negatively charged and has a diameter of 8 mm, is attached to a device that suctions onto the patient’s cornea using a suction ring. The second electrode, which is positively charged, is attached to a patch placed on the cervical vertebrae or forehead. The corneal ring electrode is placed on the cornea and filled with 0.5 mL of riboflavin solution. Then, a constant current generator is set at 1 mA and the molecule is transported through the epithelium within 5 min, resulting in a total dose of 5 mA/5 min. Stromal soakage with riboflavin is verified using a slit lamp, followed by a UV-A irradiation. The cornea is flushed with a balanced saline solution after the procedure, and a therapeutic contact lens is placed to protect the epithelium from potential UV damage during the first few days after the procedure [52]. Recent research has shown that there are two protocols of I-CXL that differ in duration of the procedure, with one lasting 5 min and the other 10 min. After a 12-month follow-up period, the longer technique has been found to be more effective than the 5 min procedure in slowing the advancement of KC. Compared to CCXL procedures, the I-CXL operation offers a reduction in postoperative complications including corneal edema or haze. Moreover, functional improvements occur at a faster pace with the I-CXL procedure. This includes a decrease in average keratometry values, stabilization of uncorrected astigmatism, and an increase in endothelial cell density (ECD) within 12 to 24 months [52]. According to a study conducted by Bikbova G. and Bikbov M., I-CXL has been shown to stabilize the progression of KC for up to 12 months [55]. In another study, Deshmukh R. et al. demonstrated, that there is a decrease in postoperative pain and fewer cases of infective keratitis [51]. Studies conducted ex vivo, have confirmed that the I-CXL procedure is more effective at halting the progression of KC than TE-CXL, but not as effective as the CCXL technique [56]. Overall, I-CXL is considered a satisfactory substitute for CCXL, despite evidence indicating that it results in only limited flattening [57].

## 7. Diluted Alcohol and Iontophoresis-Assisted Corneal Cross-Linking (DAI-CXL)

In 2017, Bilgihan K. et al. introduced a modified version of I-CXL named transepithelial DAI-CXL [31]. This innovative CXL protocol incorporated two enhancers: a diluted alcohol solution containing 10% ethanol and iontophoresis. The goal of this protocol was to enhance the diffusion of riboflavin into the corneal stroma by utilizing the intact epithelium. Additionally, the total UV-A dose could be increased to 7.2 J/cm^2^ to ensure an effective cross-linking within the stroma. A total of 93 eyes of 80 patients diagnosed with KC underwent treatment using either CCXL or DAI-CXL. Measurements of CDVA, keratometry, KC indexes, pachymetry, and aberrations were taken before the treatment and at 1, 3, 6, and 12 months following the procedure. A total of 46 eyes of 40 patients (comprising 21 males and 19 females) underwent the DAI-CXL procedure, while 47 eyes of 40 patients (consisting of 22 males and 18 females) underwent the CCXL technique. Following the DAI-CXL treatment, the epithelium remained undisturbed, and the therapeutic contact lenses were easily removed from all eyes on the first day after the procedure. On the other hand, within the CCXL group, epithelial healing occurred over a period of 2 to 3 days. Throughout the 12-month follow-up duration, none of the patients encountered infections, significant visual haze, or any other complications. Studies exhibited a significant improvement in visual, topographical, and aberrometric parameters within both the DAI-CXL and CCXL groups throughout the 12-month post-treatment follow-up. The authors observed a faster improvement in UCDVA and CDVA with DAI-CXL (at the 3-month mark) compared to CCXL (at the 6-month mark). Additionally, both the DAI-CXL and CCXL groups demonstrated a substantial enhancement in the corneal symmetry index following a 6-month follow-up. Moreover, exclusively the DAI-CXL group exhibited a significant enhancement in CDVA after 12 months of follow-up. The results also indicated that the demarcation line was visible in all individuals within both the DAI-CXL and CCXL groups at one-month post-procedure. However, it had completely disappeared in all these patients after 12 months of treatment. Long-term clinical outcomes have demonstrated improvements in CDVA and changes in corneal topography similar to those associated with the epithelium-off CXL method over a 4-year follow-up period [31,58].

Another study that was conducted to evaluate and compare the long-term effects of DAI-CXL and epithelium-off CXL was focused on KC patients with thin corneas (<400 μm). The research involved a total of 13 patients who underwent DAI-CXL and 12 patients who underwent ACXL. It was demonstrated that no patient exhibited a postoperative epithelial defect. In cases of ACXL treatment, the epithelial defect had healed within 3–4 days, after which the contact lens was removed. Throughout the follow-up period, none of the treated patients in either group experienced haze or scarring that could threaten vision or infections. Moreover, the median uncorrected distance visual acuity (UDVA) showed significant improvement in the ACXL group at both 12 and 24 months when compared to the initial measurements (*p* = 0.039 and *p* = 0.046, respectively). Conversely, in the DAI-CXL group, the UDVA showed improvement only at the 24-month mark (*p* = 0.028). Analysis of HOAs revealed a significant improvement solely within the DAI-CXL group at the 24-month mark (*p* = 0.004). However, there were no significant differences in the median changes in HOAs between the two groups during the 24-month follow-up assessment (*p* = 0.114) [28].

## 8. Pulsed Methods

Protocols involving the irradiation of the cornea with discrete light pulses, rather than continuous irradiation, have been developed based on experimental data revealing limitations in the applicability of the Bunsen–Roscoe law of photochemical reciprocity. Specifically, this law is found to only hold for illumination intensities not exceeding 50 mW/cm^2^ and durations not surpassing 2 min. The reduced efficacy of accelerated protocols, when compared to the conventional Dresden protocol, has been attributed to heightened oxygen consumption and subsequent depletion of oxygen species crucial for the crosslinking process in the cornea [59]. During the initial 10 to 15 s of UV-A exposure, aerobic conditions still prevail, and the photooxidation of the substrate (comprising proteoglycans and collagen) predominantly occurs through its reaction with photochemically generated reactive oxygen species (ROS), such as singlet molecular oxygen. This observation is consistent with a type II photochemical mechanism. Subsequently, beyond the first 10 to 15 s, oxygen becomes entirely depleted, and the reaction between the substrate and riboflavin shifts toward a predominantly type I photochemical mechanism. The strategic pulsing of UV light during the crosslinking treatment theoretically reinitiates the photodynamic type II reaction, effectively enhancing the oxygen concentration and facilitating a greater release of singlet oxygen for the crosslinking of collagen molecules [60].

The qualitative corneal changes and penetration of pulsed and continuous light-accelerated crosslinking were assessed in a clinical study conducted by Mazzotta C. et al., involving 20 adolescent and young adult patients. The experimental group was divided into two subgroups, with one undergoing treatment with a pulsed accelerated protocol (PL-ACXL) employing 30 mW/cm^2^ over 8 min (1 s on/1 s off), while the other group underwent treatment with a continuous accelerated protocol using 30 mW/cm^2^ over 4 min. Both protocols employed an epithelium-off approach, ultimately receiving an equivalent energy dose of 7.2 J/cm^2^. The corneal tissue after treatment was precisely analyzed using in vivo scanning laser confocal microscopy (IVCM), which revealed that PL-ACXL exhibited a marginally greater ability to penetrate deeper into the corneal stroma (200–240 μm) compared to the continuous light (CL-ACXL) treatment (160–200 μm). Notably, both protocols yielded similar outcomes concerning epithelial regrowth and nerve generation, and demonstrated a safe profile for the corneal endothelium in patients with progressive KC, with a significantly reduced treatment duration of under 20 min compared to the Dresden protocol [60]. The results from a one-year follow-up confirmed the effectiveness of both methods. However, PL-ACXL exhibited a slightly superior functional outcome, both in UDVA and CDVA, although statistical significance was not observed between the two treatment modalities [61].

A prospective study conducted in 2017 by Jiang LZ. et al. aimed to compare the safety and efficacy of two crosslinking methods: PL-ACXL administered at 30 mW/cm^2^ for 8 min with a 1 s on/1 s off pattern, and CL-CXL carried out at 3 mW/cm^2^ for 30 min. The study included 72 eyes from 58 patients. The findings of this study also concluded that both methods are safe and efficient, with both groups demonstrating statistically significant improvement in CDVA and refractive outcomes during the 12-month follow-up period. Moreover, there were no statistical differences observed in the ECD between the baseline and the 12-month postoperative period. Interestingly, in contrast to the results reported by Mazzotta C. et al., the study by Jiang LZ. et al. revealed that the demarcation line depth was significantly deeper in the CL-CXL group when compared to the PL-ACXL group. Consequently, the CL-CXL technique exhibited superior visual and topographic outcomes compared to PL-ACXL. However, it is worth noting that PL-ACXL induced less microstructural damage, providing a potential advantage in terms of corneal tissue integrity. Notably, the incidence of transient corneal haze was significantly higher in the CL-CXL group when compared to the PL-ACXL group. This difference in haze formation could be attributed to the longer exposure time of the corneal stroma during the CL-CXL approach [62].

Ziaei M. et al. reported on the 2-year outcomes observed in 40 patients who underwent a transepithelial PL-ACXL approach with pulsed illumination (1 s on/1 s off) using 45 mW/cm^2^ for 5 min and 20 s, delivering a surface dose of 7.2 J/cm^2^. Overall, this approach appears to be an effective method for arresting the progression of KC over a 24-month follow-up period. It is important to note that this study has a single cohort design and lacks a control group undergoing alternative CXL methods for comparison. However, the mean depth of the demarcation line is reported to be approximately 186 ± 19 μm, which aligns with findings from other published reports following TE-ACXL. Nevertheless, it should be acknowledged that the depth of the demarcation line appears to be more superficial when compared to other protocols such as ACXL or I-CXL. As a consequence, the authors cautiously advise against the routine implementation of this crosslinking protocol. Instead, they recommend employing this method of CXL selectively in patients with thinner corneas, those who may have difficulty cooperating with the epithelium-off CXL protocol, and individuals who might face challenges adhering to postoperative treatment and follow-up [59].

In a recent study conducted by Yousif MO. et al., a larger cohort comprising 103 eyes of 62 patients underwent PL-ACXL with an irradiance of 30 mW/cm2 for 4 min, while 87 eyes of 51 patients received CL-ACXL treatment at a power of 12 mW/cm^2^ for 10 min. The study’s findings revealed statistically insignificant differences between the two groups concerning visual, refractive, and keratometric outcomes during the postoperative period. Moreover, the depth of the demarcation line was insignificantly deeper in the PL-ACXL group when compared to the CL-ACXL group [63].

The safety profile of CXL was the subject of a study performed by Abdel-Radi M. et al., wherein detailed changes in the corneal endothelium after PL-ACXL with an epithelium-off approach were reported using specular microscopy. The authors noted a slight, non-significant reduction in ECD at both 3 and 6 months after PL-ACXL, indicating overall stability in ECD. Other morphological parameters, such as hexagonal cell percentage and the average endothelial cell size, showed non-significant changes at 3 and 6 months postoperatively. The mean depth of the demarcation line at 1 month postoperatively was 214 ± 17.43 μm, denoting a good level of penetrability into the corneal stroma and aligning with findings from similar studies [64].

Hernandez-Camarena JC. et al. conducted a comparative study between two accelerated pulsed protocols to assess their visual and topographic outcomes. The protocols with irradiances of 30 mW/cm^2^ over 8 min (30*8) and 45 mW/cm^2^ over 5 min and 20 s (45*5:20) did not demonstrate superiority over each other concerning UDVA or CDVA. Furthermore, there were no significant changes observed in ECD during the follow-up, and between the two groups (*p* > 0.05). The study also revealed that there was no statistically significant correlation between the depth of the demarcation line and changes in topographic astigmatism. However, when comparing the two different protocols, a statistically significant difference in the depth of the demarcation line was observed. The depth was measured at 200.63 ± 10.01 μm for the 45*5:20 cohort and 184.53 ± 19.68 μm for the 30*8 cohort, respectively (*p* < 0.001). Despite this observation, no statistically significant differences were found in UDVA or CDVA when comparing the two groups throughout the follow-up period, and the hypothesis of a correlation between the demarcation line depth and postoperative visual and topographic outcomes does not hold. At the 12-month follow-up, both protocols appeared to be safe and effective in the treatment of KC. However, it is essential to note that the study’s limitations, such as the small sample size, the absence of a control group undergoing the conventional Dresden protocol, the retrospective and comparative nature of the study (without blinding or randomization), and the possible correlations of data obtained from patients having treatment on both eyes, may introduce potential sources of bias [65].

## 9. Slit-Lamp CXL

Slit-lamp CXL is a method first described by Hafezi et al. [36]. In that method, topical anesthesia is applied 10 min before the procedure. Eyelids and surrounding areas are disinfected with chloramphenicol. Epithelial debridement is conducted using a 40% ethanol-soaked cotton swab with tapping for 70 s and gentle pressure. Riboflavin is applied while the patient is supine for 10 min. Ultrasound pachymetry measures stromal thickness after 5 and 10 min. The patient receives ultraviolet irradiation after returning to the slit lamp, all without the need for an operating room with lower costs. The depth of the demarcation line after seated slit lamp-based accelerated epithelium-off corneal cross-linking (9 mW/cm^2^ for 10 min) was assessed to be 189.4 µm. This aligns with supine CXL data, implying patient orientation does not impact demarcation line depth during CXL [66]. There was no increased risk of infection reported in this procedure, although the safety concerns were raised by Peyman A et al. who claimed that the direct ultrasound contact or performing the procedure outside the operating room might increase the risk of infectious keratitis [67,68].

## 10. The CXL Combined Method

The CXL combined method (or CXL plus method) was developed to target visual outcomes and halt the progression of KC simultaneously. Combined methods include PRK, PTK (phototherapeutic keratectomy), the use of intrastromal ring (for example MyoRing), or corneal transplantation with concurrent or subsequent CXL. Combined treatments have the potential to improve visual outcomes but require expensive equipment. Further research is needed for the widespread adoption of these methods into clinical practice.

Myoring implantation is a surgical procedure with proven clinical outcomes. The surgical procedure involves creating a pocket within the corneal stroma using a microkeratome or femtosecond laser. The pocket is closed along its entire circumference, except for a small incision tunnel. A ring implant is then inserted into the pocket. The implant is made of poly (methyl methacrylate) and has specific dimensions based on the desired refractive correction. The implantation is performed using a special applicator and is visible to the surgeon through a transparent applicator. The procedure acts by changing the curvature and shape of the cornea [32]. In a study by Daxer A., after the procedure, there was a significant improvement in UDVA by nearly 10 lines, in CDVA, which improved by almost three lines, the keratometry readings decreased by an average of 5.76 D, manifest spherical and cylindrical refractive errors, and SE [69]. The study by Bikbova et al. compared two procedures: 41 eyes after MyoRing implantation and 39 eyes after the combined procedure: MyoRing implantation and CXL. Both procedures have shown effectiveness in treating KC. After 36 months, slightly better outcomes in SE and keratometry readings were observed in the group that underwent MyoRing combined with CXL [70].

The Athens protocol, introduced by Krueger R. and Kanellopoulos A., involves a sequential approach comprising manual epithelial debridement, and partial refractive correction, which corrects approximately 70% of both cylinder and sphere measurements, with a maximum ablation depth of 50 μm through topography-guided PRK. Following the PRK procedure, 0.02% mitomycin C is applied to inhibit cell proliferation and reduce the risk of corneal haze formation. Subsequently, CCXL is performed [71]. In a Kanellopoulos A. study, there were 144 eyes examined in 10 years. The study showed significant improvements in UDVA, CDVA, corneal thickness, steep keratometry, and Kmax. Ectasia stabilization was observed in 94.4% of cases, with a minority experiencing overcorrection or hyperopic shift [33].

The Cretan protocol, developed by Kymionis G. et al., offers an alternative combined protocol with the CXL technique and transepithelial PTK (t-PTK). The depth of ablation is 50 μm within a 7.0 mm zone. Mechanical debridement is then used to enlarge the de-epithelialized area until it reaches an 8.0 mm diameter. T-PKT is followed by CCXL [72]. In the study by Grentzelos M. et al., 15 eyes of 13 patients underwent a procedure according to the Cretan protocol. The control group consisted of 15 eyes of 13 patients who underwent CCXL. The Cretan protocol group showed significant improvements in CDVA and corneal astigmatism, while the Dresden protocol group did not show significant improvements. Epithelial removal using t-PTK during CXL demonstrated better long-term outcomes compared to mechanical epithelial debridement in a four-year follow-up [73]. The Cretan protocol plus is an extension of the Cretan protocol, where after topical anesthesia, the corneal epithelium undergoes t-PTK. This involves ablation of a 6.5 mm zone at 50 mm depth, followed by mechanical enlargement to 8.0–9.0 mm. Subsequently, conventional PRK is performed with a maximum depth of 50 mm and a zone of 5.5 mm. Corneal thickness is monitored, and riboflavin is applied accordingly. UV-A irradiation followed for 30 min with an intended irradiance of 3.0 mW/cm^2^ [74]. In the study by Grentzelos M. et al., 31 eyes of 22 patients underwent simultaneous t-PTK and conventional PRK followed by CXL using the Cretan protocol plus. No complications during the procedure were observed, and significant improvements were noted in CDVA, refractive error, and corneal astigmatism up to the 3-year follow-up. ECD remained stable throughout the study period [34].

The Tel Aviv Protocol (ePRK-CXL) is a modified version of the Cretan protocol, where the epithelium is removed using an excimer laser. It includes correction of 50% of the refractive astigmatism and allows for a maximum ablation depth of 50 μm, followed by additional CXL [75]. The study by Rabina G et al. included 131 eyes, divided into the Tel Aviv Protocol group (50 patients) and the alcohol-assisted epithelial removal (AlCCXL) group (81 patients). The Tel Aviv Protocol group showed significant improvements in UDVA, CDVA, Kmax, and cylinder compared to non-significant changes in the AlCCXL group. In a study by Kaiserman I. et al. of 20 eyes, it was found that the Tel Aviv Protocol resulted in a significant improvement in UCVA and a reduction in Kmax by 2.21 D. Importantly, this was achieved with less tissue ablation compared to the Athens protocol (46 vs. 70 μm) [35].

## 11. Conclusions

The evolution of CXL techniques reflects significant advancements in the understanding and application of this treatment modality for corneal ectatic disorders. Several directions of evolution include enhanced customization and precision (topography-guided and pulsed protocols), improved safety and patient comfort (epi-on CXL, ACXL, I-CXL), time efficiency without compromising effectiveness (ACXL), increased understanding of corneal biomechanics (PL-CXL), and integration with other surgical procedures, such as PKT, PRK, or corneal ring implantation.

The evolution of CXL methods demonstrates a consistent drive towards safer, faster, and more effective treatments. While the original Dresden protocol laid a solid foundation, the field is moving towards more sophisticated approaches that integrate technological advances, an improved understanding of corneal biomechanics, and patient-specific customization. These developments not only enhance the scope of CXL but also underline the importance of ongoing research to refine and expand its applications in corneal pathology.

## Figures and Tables

**Figure 1 jcm-14-01702-f001:**
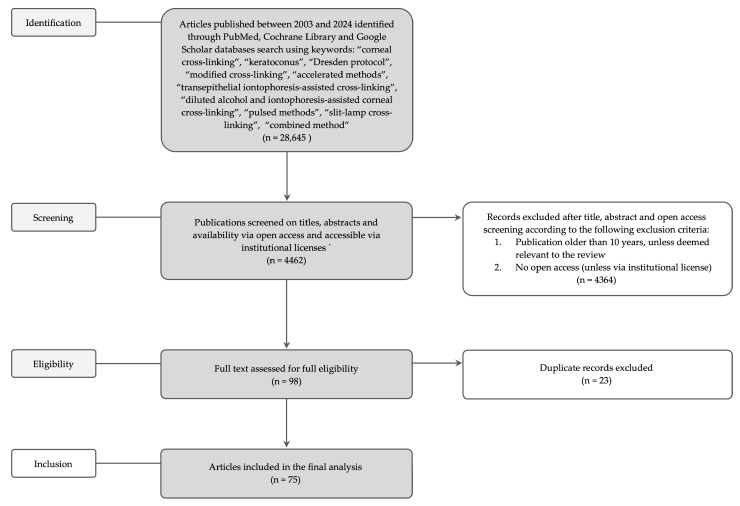
The flow chart for study selection.

**Table 1 jcm-14-01702-t001:** Summary of corneal cross-linking methods and their clinical outcomes.

Corneal Cross-Linking Method	Results
Dresden Protocol CXL	Recognized as the gold standard for treating keratoconus. The Dresden Protocol presents a 70% disease regression in clinical studies. Long-term follow ups confirm reduced keratometric values and improved visual acuity outcomes. Inhibition of keratoconus progression in 89% of patients post-procedure.
Accelerated CXL	Effective in mild to moderate keratoconus cases. Short treatment duration (3 to 10 min). Clinical outcomes, such as corneal flattening and depth of demarcation, are still variable, leading to discussions about the superiority of conventional CXL.
Transepithelial CXL	Noninvasive, low electrical current method which enhances riboflavin penetration via the corneal epithelium. Stabilization of keratoconus progression for up to 12 months. Limited corneal flattening in comparison to conventional CXL methods.
Transepithelial Iontophoresis-Assisted Crosslinking	Noninvasive method of corneal stroma penetration performed transepithelially. TE-CXL offers significant benefits such as reduction in post-operative complications, quick recovery and prompt stabilization of uncorrected astigmatism, increased endothelial cell density.
Diluted Alcohol and Iontophoresis CXL	A treatment method where riboflavin diffusion occurs transepithelially. Improvements in VA and corneal symmetry. Presentation of beneficial findings such as rapid recovery, absence of infections or haze post-procedure compared to conventional CXL.
Pulsed CXL	Irradiation of the cornea using discrete light pulses. Pulses penetrate the corneal stroma more effectively than continuous light. 12-month follow-up demonstrates similar outcomes with less microstructural damage.
Slit-lamp CXL	An effective and cost-efficient method for treating keratoconus. No increased risk of infection has been reported. However, safety concerns were reported due to the practice being performed outside the operating room.
Combined CXL methods	Advanced equipment is required. The combined CXL methods, such as MyoRing implantation and the Athens, Cretan, Tel Aviv protocols, have proven significant effectiveness in enhancing visual acuity and stabilizing keratoconus progression. The aforementioned techniques may lead to improvements in uncorrected distance visual acuity and corrected distance visual acuity. Long-term follow-ups present promising outcomes with corneal stability and reduced risk of complications.

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
