# Peer review of "A Review of Keratoconus Cross-Linking Treatment Methods"

_jcm, 2025, doi:10.3390/jcm14051702_

Round 1
Reviewer 1 Report
Comments and Suggestions for Authors
The authors present a very interesting review of corneal crosslinking (CXL). It is very well written and with an impressive analysis of the current literature. I really enjoyed reading it and I congratulate the authors.
I would only point out some minor issues:
- Dresden protocol CXL. At the beginning of the procedure, the central part of 7 mm corneal epithelium is gently rubbed to remove its outer layer. Please clarify: what is the inner layer of the epithelium? Is it the basement membrane?
- Dresden protocol CXL. The authors confirmed the safety of the method, the long-term result of reducing keratometric values (from 61.5 D to 55.3 D), and improving visual acuity (0.14 logMAR). Please clarify: mean improvement of logMAR 0.14?
- Dresden protocol CXL. CCXL represents a landmark achievement in the treatment of KC and other ectatic disorders. Please clarify: which are these other disorders?
- Dresden protocol CXL. Its use has been validated across a wide range of populations. Please clarify the populations and ethnic involvement in the aforementioned studies.
- Manuscript. Several times, the term ‘visual acuity’ appears, referring to improvements or worsening in the papers. Please use UDVA or CDVA, but not the misleading ‘visual acuity’ expression.
- Acronyms.
Please use corrected distance visual acuity (CDVA) instead of the old-fashioned BCVA and BCDVA.
Please use uncorrected distance visual acuity (UDVA) instead of the old-fashioned UCDVA.
Please use A-CXL or ACXL, but not both. Sometimes, ‘AXCL’ appears.
Please use C-CXL or CCXL, but not both.
Please use PL-A-CXL or PL-ACXL, but not both.
Author Response
Dear Reviewer,
Thank you very much for valuable comments and clues. We carefully revised our manuscript and included all reviewers’ recommendations. We appreciate your comments and clues.
Answers and comments to the Reviewer 1:
- Dresden protocol CXL. At the beginning of the procedure, the central part of 7 mm corneal epithelium is gently rubbed to remove its outer layer. Please clarify: what is the inner layer of the epithelium? Is it the basement membrane?
Thank you for the right suggestion. The sentence is misleading. The entire epithelium is removed during the procedure.
The sentence was corrected to:
“At the start of the procedure, the central part of a 7 mm area of corneal epithelium is gently rubbed until it is removed to facilitate the solution's penetration”.
- Dresden protocol CXL. The authors confirmed the safety of the method, the long-term result of reducing keratometric values (from 61.5 D to 55.3 D), and improving visual acuity (0.14 logMAR). Please clarify: mean improvement of logMAR 0.14
Thank you for your comment. According to a study by Raiskup et al. “Corneal collagen crosslinking with riboflavin and ultraviolet-A light in progressive keratoconus: Ten-year results”, there was an improvement in corrected distance visual acuity (CDVA) by 0.14 logMAR units. This improvement in logMAR corresponds to an enhancement of approximately 1.4 lines on a standard Snellen chart, signifying better visual clarity.
We corrected the sentence to:
“The authors confirmed the safety of the method, the long-term result of reducing keratometric values (from 61.5 D to 55.3 D), and improving visual acuity (0.14 logMAR units; approximately 1.4 lines on a standard Snellen chart) [9]”.
- Dresden protocol CXL. CCXL represents a landmark achievement in the treatment of KC and other ectatic disorders. Please clarify: which are these other disorders?
Thank you for your comment. This indeed needs to be clarified.
The Dresden protocol for corneal collagen cross-linking (CXL) has proven beneficial not only for keratoconus (KC) but also for other corneal ectatic disorders. These include: pellucid marginal degeneration (PMD), Post-refractive surgery ectasia, keratoglobus, Terrien's marginal degeneration, iatrogenic ectasia, and selected corneal dystrophies, such as lattice corneal dystrophy.
We added the list to the sentence:
“CCXL represents a landmark achievement in the treatment of KC and other ectatic disorders. These include: pellucid marginal degeneration (PMD), Post-refractive surgery ectasia, keratoglobus, Terrien's marginal degeneration, iatrogenic ectasia, and selected corneal dystrophies, such as lattice corneal dystrophy.”
- Dresden protocol CXL. Its use has been validated across a wide range of populations. Please clarify the populations and ethnic involvement in the aforementioned studies.
Thank you for your comment. We expanded this part of the manuscript and added the references.
We rewrote the paragraph in the manuscript:
“Its use has been validated across a wide range of populations, clinical settings, and stages of KC including European: Germany (Wollensak et al.), Italy (Caporossi et al.), France (Poli et al.), the UK (O'Brart et al., Chowdhury et al. KERALINK); North America: United States (Hersh et al.), Canada (Legare et al.); Middle East: Saudi Arabia ( Khattak et al.), Iran (Hashemi et al.); Asia: India (Sachdev et al.), China (Gu et al.); Australia (Wittig-Silva et a.); Africa (Mark et al.); South America: Brazil (Gadelha et al). [7,10,21–25,11,14–20]”.
- Several times, the term ‘visual acuity’ appears, referring to improvements or worsening in the papers. Please use UDVA or CDVA, but not the misleading ‘visual acuity’ expression.
Thank you for your comment. We corrected the manuscript according to the suggestion.
Thank you for your valuable suggestions. We implemented all requested corrections.
Please use corrected distance visual acuity (CDVA) instead of the old-fashioned BCVA and BCDVA.
Please use uncorrected distance visual acuity (UDVA) instead of the old-fashioned UCDVA.
Please use A-CXL or ACXL, but not both. Sometimes, ‘AXCL’ appears.
Please use C-CXL or CCXL, but not both.
Please use PL-A-CXL or PL-ACXL, but not both.
Reviewer 2 Report
Comments and Suggestions for Authors
1. The article deals with an important and current topic and reports all the innovative methods on corneal cross-linking. However, major revisions are needed.
2. In the abstract on line 33 “numerous others” is a generic expression. It would be better to include all the methods analyzed in the review in order of treatment.
3. The materials and methods chapter should be revised almost completely: Were the Prisma guidelines on systematic reviews followed? If yes, the inclusion and exclusion criteria, search string, Prisma flow chart, quantitative data on how many articles were included and excluded, and the reasons why (summarized in a table in the appendix) are missing.
4. The results chapter lacks a results table. This needs to be added.
5. Many contents (to give an example those contained between lines 148-154, 568-571) are in an assertive tone despite being in the results chapter. They should either be removed or included in an additional final “discussion” chapter (before the conclusions)
6. Is Chapter 4 a chapter of general introduction to modified CXL? If yes, it is unclear in its development: it is too specific about accelerated methods (first paragraph) and pulsed methods (third paragraph). It also does not include all the alternative methods to the Dresden protocol such as DAI-CXL, slit lamp methods, and combined methods. It would need to be revised in its entirety to make it a brief but effective excursus on the methods discussed later in the article.
7. There are multiple excessively long paragraphs, which are beyond the summary and punctual nature of a systematic review. To give some examples: lines 382-429, 430-461, 547-567. These paragraphs should be summarized by reporting only the highlights of the study under review (hence the additional need for a table of contents).
Author Response
Dear Reviewer,
Thank you very much for valuable comments and clues. We carefully revised our manuscript and included all reviewers’ recommendations. We appreciate your comments and clues.
Answers and comments to the Reviewer 2:
- In the abstract on line 33 “numerous others” is a generic expression. It would be better to include all the methods analyzed in the review in order of treatment.
This is a very valuable comment. We modified the sentence and listed all methods and modifications, which were analyzed.
“These modifications encompass approaches like transepithelial or epithelium-on CXL (TE-CXL or epi-on CXL), accelerated CXL (ACXL), pulsed CXL (PL-CXL), and numerous others, such as transepithelial iontophoresis-assisted crosslinking (I-CXL), diluted alcohol and iontophoresis-assisted corneal cross-linking (DAI-CXL), slit-lamp CXL, and or CXL plus (combined) methods”.
- The materials and methods chapter should be revised almost completely: Were the Prisma guidelines on systematic reviews followed? If yes, the inclusion and exclusion criteria, search string, Prisma flow chart, quantitative data on how many articles were included and excluded, and the reasons why (summarized in a table in the appendix) are missing.
Thank you for your suggestions. We agree with your comment. We implemented the Prisma flow chart to the manuscript.
- The results chapter lacks a results table. This needs to be added.
Thank you for your comment. We added the table with the summary of the CXL protocols.
- Many contents (to give an example those contained between lines 148-154, 568-571) are in an assertive tone despite being in the results chapter. They should either be removed or included in an additional final “discussion” chapter (before the conclusions)
Thank you for your suggestions. We removed the suggested sections to avoid an assertive tone. We aimed to analyze the results objectively.
- Is Chapter 4 a chapter of general introduction to modified CXL? If yes, it is unclear in its development: it is too specific about accelerated methods (first paragraph) and pulsed methods (third paragraph). It also does not include all the alternative methods to the Dresden protocol such as DAI-CXL, slit lamp methods, and combined methods. It would need to be revised in its entirety to make it a brief but effective excursus on the methods discussed later in the article.
Thank you for your insightful feedback. We have thoroughly revised Chapter 4 to provide a cohesive and comprehensive introduction to modified CXL techniques. The updated section now offers a balanced overview of all relevant methods, including DAI-CXL, slit-lamp delivery techniques, and combined approaches, ensuring it serves as an effective prelude to the detailed discussions in the subsequent sections. We appreciate your thoughtful suggestions, which have significantly contributed to improving the structure and clarity of this chapter.
- There are multiple excessively long paragraphs, which are beyond the summary and punctual nature of a systematic review. To give some examples: lines 382-429, 430-461, 547-567. These paragraphs should be summarized by reporting only the highlights of the study under review (hence the additional need for a table of contents).
Thank you for a valuable comment. We have carefully revised the identified sections, condensing the content to focus on the key highlights and results of the studies.
Round 2
Reviewer 2 Report
Comments and Suggestions for Authors
- Revisions 1--3-5-6-7 have been adequately corrected.
- Revision 2 is incomplete with the appendix regarding the exclusion of screened studies as full text.
- Revision 4 is not acceptable because the results table is still missing. By results table is meant, as indicated by the Prisma Flow Charts, an inclusive table of all the papers included in the review reported with key parameters (e.g., collateral effects, efficacy parameters, individual protocol methods...)
Author Response
We greatly appreciate the reviewer's insightful comments, which have further guided us in enhancing the quality of the manuscript during this second round of revisions.
Revision 2 is incomplete with the appendix regarding the exclusion of screened studies as full text.
Thank you for the comment. Lines 102-107, we provided the exclusion criteria:
"Records were excluded based on title, abstract, and open access screening according to the following criteria: publications older than 10 years (unless relevant to the review) or lack of open access unless they were accessible via institutional licenses. Duplicate records were also removed, resulting in a final selection of 75 articles for analysis. The flow chart for study selection is presented in Figure 1". We also corrected the figure 1.
Revision 4 is not acceptable because the results table is still missing. By results table is meant, as indicated by the Prisma Flow Charts, an inclusive table of all the papers included in the review reported with key parameters (e.g., collateral effects, efficacy parameters, individual protocol methods...)
Thank you. We provided a table summarizing all the papers included in the review with key parameters in the supplementary files.